# Optimising Patient Outcomes in Tongue Cancer: A Multidisciplinary Approach

**DOI:** 10.3390/cancers16071277

**Published:** 2024-03-26

**Authors:** Jasper de Boer, Rebecca Barnett, Anthony Cardin, Michelle Cimoli, Lauren Davies, Clare Delany, Benjamin J. Dixon, Sue M. Evans, Michael W. Findlay, Carly Fox, Maria Ftanou, Christopher D. Hart, Megan Howard, Tim A. Iseli, Andrea Jackson, Sevastjan Kranz, Brian H. Le, Ernest Lekgabe, Rachel Lennox, Luke S. McLean, Paul J. Neeson, Sweet Ping Ng, Lorraine A. O’Reilly, Anand Ramakrishnan, David Rowe, Carrie Service, Ankur Singh, Alesha A. Thai, Albert Tiong, Tami Yap, David Wiesenfeld

**Affiliations:** 1VCCC Alliance, Melbourne 3052, Australia; jasper.deboer@hudson.org.au; 2St. Vincent’s Hospital Melbourne, Melbourne 3065, Australia; rebecca.barnett@svha.org.au (R.B.); christopher.hart@svha.org.au (C.D.H.); andrea.jackson@svha.org.au (A.J.); 3Peter MacCallum Cancer Centre, Melbourne 3000, Australiac.delany@unimelb.edu.au (C.D.); mifindlay@mac.com (M.W.F.); maria.ftanou@petermac.org (M.F.); megan.howard@petermac.org (M.H.); brian.le@mh.org.au (B.H.L.); alesha.thai@petermac.org (A.A.T.); albert.tiong@petermac.org (A.T.); 4Austin Health, Heidelberg 3084, Australia; michelle.cimoli@austin.org.au (M.C.); rachel.lennox1@gmail.com (R.L.); sweetping.ng@austin.org.au (S.P.N.); 5The Royal Melbourne Hospital, Melbourne 3052, Australia; lauren.davies@mh.org.au (L.D.); carly.fox@mh.org.au (C.F.); tim.iseli@mh.org.au (T.A.I.); sevastjan.kranz@mh.org.au (S.K.); ernest.lekgabe@mh.org.au (E.L.); anand.ramakrishnan@mh.org.au (A.R.); carrie.service@mh.org.au (C.S.); tspyap@unimelb.edu.au (T.Y.); 6Department of Surgery, University of Melbourne, Melbourne 3010, Australia; 7Victorian Cancer Registry, Cancer Council Victoria School of Public Health and Preventive Medicine, Monash University, Melbourne 3002, Australia; sue.evans@cancervic.org.au; 8The Sir Peter MacCallum Department of Oncology, University of Melbourne, Melbourne 3010, Australia; paul.neeson@petermac.org; 9The Walter and Eliza Hall of Medical Research (WEHI), Melbourne 3052, Australia; oreilly@wehi.edu.au; 10Centre for Epidemiology and Biostatistics, University of Melbourne, Melbourne 3010, Australia; ankur.singh@unimelb.edu.au; 11Melbourne Dental School, University of Melbourne, Melbourne 3010, Australia

**Keywords:** tongue neoplasms, multidisciplinary care, oncologic rehabilitation

## Abstract

**Simple Summary:**

The initiative outlined in this paper emphasises the importance of a collaborative strategy in managing tongue cancer, involving a broad spectrum of healthcare professionals. This multidisciplinary method integrates the expertise of nurses, allied health workers, and various medical specialists to enhance diagnosis, treatment, and recovery processes for patients. By developing and examining a series of online lectures, this study highlights the critical roles played by each discipline in the comprehensive care pathway, from prevention and early detection to the management of advanced stages of the disease. This approach aims not only to address the clinical aspects of tongue cancer but also to support the overall well-being of patients, showcasing the benefits of teamwork in healthcare settings for improving patient outcomes.

**Abstract:**

A multidisciplinary approach to the management of tongue cancer is vital for achieving optimal patient outcomes. Nursing and allied health professionals play essential roles within the team. We developed symposia comprising a series of online lectures offering a detailed perspective on the role each discipline and consumer perspective has in the management of patients with tongue cancer. The topics, including epidemiology and prevention, diagnosis, treatment planning, surgery, adjuvant care, and the management of recurrent or metastatic disease, were thoroughly examined. The symposia highlighted the significance of fostering collaboration and continuous learning through a multidisciplinary approach. This initiative should be relevant to healthcare professionals, researchers, and policymakers striving to enhance patient outcomes in tongue cancer care through innovative collaboration.

## 1. Introduction

Squamous cell carcinoma of the tongue is a complex disease that requires a multidisciplinary approach to achieve optimal patient outcomes. The established goals of a multidisciplinary approach are to draw on the expertise of different specialties to make evidence-based recommendations for individual patient management [1]. Collaboration is essential to ensure that patients receive comprehensive care that addresses their treatment needs. The purpose of this educational review paper is to describe the importance and process of multidisciplinary tongue cancer care from diagnosis to treatment and rehabilitation, with a focus on the disciplines involved.

This paper was developed through a series of multidisciplinary dialogues involving experts from Melbourne, Australia. These dialogues were the foundation for the creation of online lectures, each tailored to the presenter’s respective specialty. The lectures then provided a more in-depth look at each field, contributing to the overall goal of enhancing patient outcomes. This process ensured a comprehensive and interconnected approach by incorporating insights into all aspects of prevention of tongue cancer and patient care. We examined the role of each specialty involved in the care of patients with tongue cancer, including epidemiology and aetiology, prevention, presentation, diagnosis, treatment planning, surgery, reconstruction, systemic therapies, radiation oncology, supportive care, survivorship, surveillance, and managing recurrent, residual, or metastatic disease including palliative care. We also highlight the critical role nursing and allied health professionals play in each of these areas.

## 2. Materials and Methods

### 2.1. Assembly of Multidisciplinary Expert Panel

A multidisciplinary expert panel was convened, composed of professionals from diverse domains and staff members of the Victorian Comprehensive Cancer Care Alliance. The panel members were chosen on the basis of their extensive experience and proficiency within their individual disciplines, aiming to promote an all-encompassing and synergistic strategy to enhance patient outcomes.

The panel comprised experts from a range of fields: epidemiology; allied health, inclusive of clinical psychology, dietetics, physiotherapy, and speech pathology; dentistry, encompassing oral medicine, prosthodontics, and special needs dentistry; medicine, spanning medical oncology, medical ethics, nuclear medicine, palliative care, pathology, psychiatry, radiation oncology, and radiology; nursing, involving nurse care coordinators and personnel from ward and operating theatres; and surgery, inclusive of oral and maxillofacial, otolaryngology head and neck, and plastic and reconstructive surgery.

### 2.2. Multidisciplinary Discussions and Literature Review

The expert panel initially convened for a series of multidisciplinary discussions. These sessions aimed at integrating the diverse perspectives and expertise of the panel members to identify areas of potential improvement in patient care, assess novel interventions, and establish and define best practices. In the literature review and data extraction phase, each expert group within the multidisciplinary panel selected publications relevant to their field. The selection criteria included publication recency, impact factor, and relevance to topics such as epidemiology, diagnosis, treatment planning, and management of tongue cancer. The groups then utilised the selected literature to extract insights, methodologies, and outcomes relevant to multidisciplinary care. This process informed further discussions and the development of online lectures, ensuring that they reflected the current practices and knowledge in the field. The discussions were structured to ensure that all aspects of patient care were considered, fostering a holistic approach to the improvement of patient outcomes.

### 2.3. Development of Online Lectures

Following the multidisciplinary discussions, each expert was tasked with the creation of an online lecture covering the core concepts and recent advancements in their specific domain. These lectures were designed to further disseminate the insights from the discussions and serve as a foundation for ongoing collaboration and knowledge exchange among the panel members. The lectures can be accessed online at the following link: https://www.youtube.com/playlist?list=PLh8-r-u9s-7FKHP328aJh5xWg9ttImJUG (accessed on 2 April 2023).

### 2.4. Synthesis of Findings

The outcomes of the multidisciplinary discussions and the development of the online lectures were systematically reviewed, analysed, and co-ordinated by the expert panel. The resulting insights and recommendations were then consolidated, providing a comprehensive and integrated approach to enhancing patient care across the various specialties.

## 3. Results

### 3.1. Epidemiology and Aetiology of Tongue Cancer

In 2022, Australia recorded 1304 cases of tongue cancer, predominantly in males (945 cases), with an incidence rate of 4.4 per 100,000 population [2]. The key risk factors include chronic inflammation and tobacco, alcohol, and betel nut consumption [3]. Notably, despite national declines in tobacco and alcohol use, the tongue cancer incidence continues to rise (Figure 1), contrasting with a steady or declining mortality rate [4] (Figure 2). This trend highlights the need for further research to understand the disease’s drivers in Australia, particularly in prevention and treatment strategies. Tobacco and alcohol remain primary risk factors [5], yet a significant number of cases occur without these known risks [6]. The potential role of HPV in tongue cancer, especially relevant for indigenous Australians with higher oral HPV rates, is an area of active investigation [7], suggesting new directions for understanding and managing the disease. Additionally, the role of emerging risk factors such as smokeless tobacco use in migrant Australians and its impact on future trends must be considered carefully.

### 3.2. Applying the Optimal Care Pathway to Tongue Cancer

The Optimal Care Pathway for head and neck cancers, inclusive of tongue cancer, represents a holistic model ensuring high-quality, evidence-based patient care. This pathway begins with prevention and early detection initiatives, emphasising lifestyle changes, routine screenings, and prompt symptom evaluation [8]. Upon symptom presentation, swift diagnostic procedures and referrals to a multidisciplinary team (MDT) are crucial.

The MDT rigorously assesses cancer staging through imaging and proposes patient-specific treatment plans, considering the patient’s health status and preferences. The treatment options—surgery, radiation, chemotherapy, or their combinations—are selected based on disease stage and individual factors, supplemented by rehabilitation, speech therapy, and nutritional guidance. Supportive care addressing the patient’s physical, emotional, and practical needs is a cornerstone during and after treatment.

Post-treatment, regular follow-ups are essential for recurrence monitoring, rehabilitation, managing long-term effects, and ongoing support, including psychosocial and lifestyle interventions. For advanced cases, the early integration of patient-centred palliative care is vital, focusing on symptom management, emotional support, and end-of-life care planning. These principles of the Optimal Care Pathway are reflected throughout the subsequent sections of this paper.

### 3.3. Disease Presentation, Referral Patterns, and COVID-19

Tongue cancer typically presents as a non-healing ulcer or sore on the tongue, often accompanied by difficulty swallowing, pain, and speech changes. Oral potentially malignant disorders (OPMDs), presenting as keratotic or erythematous oral mucosal changes, precede some oral cancers [9]. Routine oral mucosal screenings are crucial for early detection, especially in at-risk groups. Referral to an MDT is recommended for suspected cases, ensuring comprehensive care covering all necessary specialties.

The COVID-19 pandemic significantly disrupted the presentation and referral patterns for tongue cancer in Victoria and Australia. A reduction in the incidence of head and neck cancers was seen among males in 2020, potentially resulting from decreased routine screenings and pandemic-related medical (including dental) attention delays. While no rebound effect was seen, the age-standardised incidence rates in 2021 and 2022 were similar to those reported pre-2020 [10].

### 3.4. Diagnosis, Staging, and Treatment Planning

The diagnosis of tongue cancer involves clinical examinations, imaging, and pathology, including a biopsy for a definitive diagnosis and, in patients with unresectable or metastatic disease, a PD-L1 combined positive score (CPS) to guide systemic treatment selection with immune checkpoint inhibitors. Staging is based on tumour size, depth of invasion, and the presence of metastasis, using the preferred imaging techniques of CT, MRI, ultrasound, and PET [11]. Treatment options range from surgery, radiation, and chemotherapy to palliative care, necessitating collaboration among medical and allied health professionals for optimal patient care.

### 3.5. Multidisciplinary Meetings

MDT meetings, now recommended by the National Comprehensive Cancer Network (NCCN) for head and neck cancers, are critical for comprehensive care and improved patient outcomes. These meetings enable collective decision-making, clinical trial discussions, palliative care considerations, and holistic patient support. An effective MDT includes various surgical specialists, oncologists, radiologists, dentists, and allied health professionals. Successful MDTs require clear communication, mutual respect, and structured meetings. Tongue cancer care raises complex ethical questions related to informed consent, patient representation, and treatment risks and benefits. Clinical ethics support is crucial, with frameworks like Jonsen, Siegler, and Winslade’s 4-quadrant model aiding teams in navigating these issues [12]. This model integrates clinical evidence with patient preferences and values. Additionally, the ethics of teamwork, led by senior clinicians, focuses on inclusive environments and aligning clinical decisions with broader patient interests and values [13].

### 3.6. Preparing the Patient for Surgery

Surgery is a primary treatment for most tongue cancer patients. The head and neck coordinator is pivotal, managing referrals, organising multidisciplinary meetings, and ensuring comprehensive patient evaluations and consultation for optimal treatment planning. This multifaceted approach addresses all care aspects, including educational and supportive needs [14].

### 3.7. Supportive Care

The Optimal Care Pathway for head and neck cancer highlights the need for supportive care throughout all stages of cancer: diagnosis, treatment, survivorship, and end-of-life care [8]. Supportive care may be categorised into five major domains: informational, emotional, psychosocial, spiritual, and physical [8]. A variety of health care professionals including medical, nursing, and allied health professionals can be involved in meeting these supportive care needs. Meeting a person’s informational supportive care needs is essential to alleviate confusion, anxiety, and fear, enabling patients and their families to participate in care discussions and decisions [14]. The coordinator’s role extends to pre- and post-surgery education, employing tools like the distress thermometer to assess and address psychological states, which are crucial for reducing anxiety and improving outcomes.

Speech pathologists and dietitians also play vital roles. Speech pathologists provide guidance on post-surgery changes in speech and swallowing functions, while dietitians focus on nutritional optimisation and managing potential malnutrition, a significant risk factor for poorer outcomes [15]. Nutrition screening should be initiated at diagnosis, with those at risk referred to dietitians. Prehabilitation aims to optimise patients before surgery and radiotherapy, considering individual risk levels. MDTs weigh gastrostomy usage and insertion timing based on key risk factors.

Including caregivers in the counselling process enhances patient support during recovery. The distress thermometer helps tailor sessions to patient needs and identifies the necessity for psychological counselling [16]. An MDT approach, with the head and neck coordinator and allied health professionals at the forefront, is vital for favourable post-operative outcomes.

### 3.8. Surgical Principles

The primary treatment for tongue cancer is surgery, aiming to remove the tumour and achieve a 5 mm histologic margin, termed a “clear” margin, around the tumour [17]. This involves the removal of affected lymph nodes in the neck and, when necessary, reconstruction with tissues to optimise function. Pre-operative planning includes assessing the anticipated defect, identifying suitable tissues for reconstruction, and evaluating patient fitness for the procedure.

### 3.9. Management of the Primary and Neck Nodes

Tongue cancer treatment demands a multidisciplinary approach, with the surgical goal of clear histopathologic margins (5 mm normal tissue around the cancer) being crucial for patient prognosis [17]. Despite efforts, suboptimal histologic margins are reported in 30 to 85% of oral SCC resections, with close or positive margins linked to increased loco-regional failure and lower survival rates, and possibly necessitating additional treatments like re-resection or adjuvant therapy [17].

Factors influencing surgical margins include discrepancies between macroscopic and microscopic tumour extent, unrecognised second primary SCCs, and tissue shrinkage post-resection. The NCCN guidelines recommend resecting 10 to 15 millimetres beyond visible and palpable cancer to achieve clear margins. High-quality pre-operative imaging and high-volume surgical teams using intraoperative adjuncts can aid in achieving clear margins.

Surgical approaches vary based on tumour size and location, ranging from partial glossectomy for smaller tumours to total glossectomy for larger ones, potentially requiring a mandibulotomy for access. Neck management includes assessing lymph node involvement, with neck dissection or sentinel lymph node biopsy as options.

### 3.10. Management of the Neck

Neck management is vital in tongue cancer treatment due to possible lymphatic spread to neck nodes. The assessment includes palpation and imaging (ultrasound, CT, MRI, PET) with fine needle aspiration biopsy for suspicious lymph nodes.

Surgical strategies for neck disease range from elective selective neck dissection and removing specific nodal basins as a staging procedure to therapeutic comprehensive neck dissection for known metastatic lymph nodes. Elective neck dissection is increasingly incorporated into treatment protocols, even for small primary tumours, due to evidence of a survival benefit. Complications can include scar formation, numbness, marginal mandibular nerve palsy, lymphatic fluid leaks, and shoulder dysfunction due to nerve injuries [18].

### 3.11. Dealing with Treatment Failure

Oral tongue cancer presents unique challenges when standard care fails, raising questions about predicting failure and tailoring care. Achieving adequate histological margins in initial resections is crucial for reducing loco-regional failure and improving survival. This includes ensure clean margins to minimise recurrence, accurately staging and managing neck metastases.

Particular patient groups, such as non-smoking, non-drinking elderly females, face higher loco-regional failure rates [19]. A study indicates worse outcomes for non-smoking, non-drinking elderly females compared to traditional smoker/drinker patients, even when matched for cancer stage [19]. Young patients, though having better overall survival, show higher rates of local-regional failure. Additionally, distance from tertiary referral centres is an independent risk factor in Victoria, with patients over 200 kilometres away experiencing poorer outcomes [20].

Elective neck dissection is advised for tumours with over 3 mm depth of invasion potentially offering a survival advantage [21]. Salvage surgery, often combined with adjuvant therapies, presents challenges, particularly for patients previously treated with radiation. The difficulty in managing oral tongue cancer lies in predicting which patients will not respond to standard care. Despite the uncertainties in managing surgical failures, these insights underscore the necessity of individualised care strategies for patients at elevated risk.

### 3.12. Reconstruction Techniques

The goal of tongue cancer reconstruction is to restore both form and function, a complex task due to the tongue’s unique structure and physiology. The choice of reconstruction technique varies based on the defect’s size and location, as well as the patient’s overall health. Options include primary and secondary closure, grafts, loco-regional flaps, and free flaps. However, the absence of a standardised classification system for tongue defects complicates outcome comparisons.

Tongue reconstruction, one of the most intricate procedures in head and neck surgery, hinges on balancing functional preservation with achieving clear surgical margins. The tongue’s complex musculature and neurovascular supply necessitate meticulous planning. Reconstruction aims to seal the oral cavity from the neck, protect the airway, and restore speech and swallow functions.

For small defects, primary closure or secondary intention healing may suffice, while larger defects often require flap reconstruction. Local flaps are options for some cases, but free flaps are more reliable. Hemiglossectomy defects might be addressed with thin fasciocutaneous flaps, whereas subtotal glossectomies demand bulkier flaps for functional effectiveness. Total glossectomies present significant challenges, requiring tailored flap approaches depending on the extent of resection to facilitate airway protection [22].

### 3.13. Post-Surgical Recovery, Rehabilitation and Functional Outcomes

Functional outcomes such as speech and swallowing are vital success indicators for surgery thus necessitating a multidisciplinary approach that incorporates speech therapy, swallowing therapy, and physiotherapy [23]. Recovery following surgery with free flap reconstruction initially focuses on re-establishing airway patency and achieving adequate swallowing function for tracheostomy decannulation. An unpublished audit of oral tongue cancer cases at St Vincent’s Hospital Melbourne and The Royal Melbourne Hospital highlighted that larger resections necessitate longer tracheostomy periods, with most patients requiring modified diets and behavioural strategies to manage dysphagia on discharge. The extent of resection directly impacts recovery duration with dysphagia severity increasing with resections extending into the tongue base, impacting pharyngeal clearance and increasing the risk of aspiration [24].

Favourable speech outcomes correlate with tongue tip preservation, smaller excision sizes, and minimal tongue tethering post-reconstruction [24]. Changes in speech intelligibility significantly impact social participation and identity [25]. Patients’ perceptions of their speech may differ from clinical assessments, highlighting the importance of patient-reported outcome measures in capturing patient perspectives on post-treatment quality of life [26].

Post-surgical physiotherapy focuses on preventing pulmonary complications, mobilising patients, and addressing musculoskeletal issues and lymphoedema [27,28,29]. Factors such as surgery duration, tracheostomy, smoking history, and respiratory comorbidities contribute to the risk of post-operative pulmonary complications (PPCs) [30]. Early mobilisation and chest physiotherapy are recommended to reduce PPCs and shorten hospital stays [29]. Additionally, physiotherapists assess and treat musculoskeletal problems like shoulder dysfunction, donor site pain, and lymphoedema, which are essential for improving quality of life (QoL) and functional recovery [27,31].

### 3.14. Systemic Therapies in Tongue Cancer

Systemic therapies such as concurrent chemotherapy with post-operative radiotherapy in the presence of extranodal extension or positive margin is integral in tongue cancer treatment. In the unresectable or metastatic setting, the role of immune checkpoint inhibitors was first examined in the second line or beyond setting.

A study comparing nivolumab, an PD-1 antibody, with cytotoxic chemotherapy in the second-line treatment for platinum-refractory metastatic or recurrent head and neck squamous cell carcinoma (SCC) showed nivolumab’s superiority in response rates and overall survival [32]. Nivolumab was more effective in patients with PD-L1 expression >1% [32]. Subsequently, the role of ICIs was examined in the first line setting in the KEYNOTE-048 study. KEYNOTE-048 established pembrolizumab, alone or combined with chemotherapy, as a preferred option for metastatic/recurrent head and neck SCC cases with treatment guided by PD-L1 expression by immunohistochemistry and quantified into a combined positive score (CPS) and clinical factors, such as tumour burden and performance status [33]. This study revealed pembrolizumab’s advantage in survival rates over the traditional EXTREME regimen, particularly in patients with high PD-L1 expression (CPS > 20).

Ongoing phase III trials are examining immunotherapy’s role in earlier disease stages, although results like those from the JAVELIN 100 and KEYNOTE-412 trials, which examined the addition of ICIs to definitive chemoradiotherapy and for 12 months following the completion of chemoradiotherapy, have not shown statistically significant improvements in progression-free survival or event-free survival [34,35]. The potential of neoadjuvant immunotherapy, with or without adjuvant therapy, is also being investigated [36].

### 3.15. Radiation Oncology in Oral Tongue Cancer Treatment

Radiotherapy is primarily used as an adjuvant therapy in potentially curable cases. It is typically administered after surgery for patients with high-risk disease features. In advanced disease when surgery is not feasible, definitive (chemo)radiation is considered for an attempted cure. For incurable patients, radiotherapy can effectively palliate symptoms including pain and bleeding.

The combination of surgery and radiotherapy evolved empirically to improve control rates, as each modality alone showed poor local and regional control [37]. A study by Peters et al. demonstrated that high-risk disease benefited from higher radiation doses [37]. Furthermore, randomised trials have indicated that patients with extra-nodal extension and positive margins benefit most from concurrent chemo-radiotherapy, reducing loco-regional recurrence, disease relapse, and mortality risk [38].

Guidelines suggest considering adjuvant radiotherapy for factors like advanced disease (T3/T4), perineural and lympho-vascular invasion, and nodal involvement [39]. The indications for radiotherapy sometimes lack consensus, often relying on retrospective data and expert opinion due to a lack of randomised trials. The necessity of adjuvant radiotherapy in early nodal neck disease remains controversial, with guidelines from organisations like ASCO not reaching a strong consensus [40].

Even in cases with expected good outcomes (early-stage disease with no neck involvement), risks persist, as highlighted by Ganly et al.’s analysis showing lower-than-expected regional recurrence-free survival rates in T1N0 and T2N0 disease [41].

### 3.16. Supportive Care and Survivorship

Post-treatment care for tongue cancer is crucial for patients’ physical health, quality of life, and social engagement. Supportive care adapts to the patient’s changing needs throughout the treatment and recovery phases. Speech pathologists and dietitians are vital in rehabilitating patients facing speech or swallowing difficulties post-treatment. Nurses play a critical role in managing the recovery of patients undergoing chemotherapy and radiation, including lymphoedema management, which is often under-recognised and undertreated [42].

Head and neck lymphoedema, which can significantly impair swallowing and speech functions, varies widely in incidence [42]. Its management involves complete decongestive therapy (CDT), encompassing education, skin care, exercise, massage, and compression [31].

Oncology specialist nurses/nurse consultants participate in multidisciplinary meetings for treatment planning, provide patient guidance on chemotherapy, including potential side effects, cytotoxic precautions, and infection risks, along with vaccination advice pre-treatment. They also coordinate referrals for audiology, pathology, fertility preservation, and oversee ongoing care and side effect management.

Radiation oncology nurses support patients starting from the initial radiation oncology consultations and through treatment. They focus on education, post-operative wound care, management of the acute side effects of radiation, and emotional support. Post-treatment, they manage care at home, recognising that side effects may worsen, and focus on survivorship, which is informed by the current research.

Addressing sexuality and sexual dysfunction is also crucial for holistic patient care. Open communication about sexuality, facilitated by healthcare professionals, and referrals to psychologists, sexologists, or psychosexual therapists can significantly improve patient well-being and quality of life.

The recovery trajectory for tongue cancer patients encompasses multiple stages, each with distinct supportive care needs [43]. Patients often experience a range of emotions and challenges, including uncertainty, life disruption, and isolation [44]. Adapting to changes in speech and swallowing becomes central to recovery, which is guided by the biopsychosocial model.

### 3.17. Support Groups

The long-term effects of head and neck cancer diagnosis and treatment can lead to QoL deterioration and feelings of isolation. Peer support groups provide valuable emotional and practical support, fostering coping, reducing distress, and enhancing self-esteem [45]. These groups cater to specific needs, such as those of female patients, who may face different challenges.

### 3.18. Managing Recurrent, Residual, or Metastatic Disease and Palliative Care

In cases of recurrent, residual, or metastatic tongue cancer, a multidisciplinary approach is essential. Advanced disease management and palliative care discussions focus on treatment options, including immunotherapy and chemotherapy. Discussions should also cover treatment benefits, QoL considerations, and toxicities. The early involvement of palliative care services is crucial to optimise the QoL in advanced disease stages.

### 3.19. Future Research Outlook

Research into early diagnoses leverages novel non-invasive imaging technologies and fluorescence agents [46]. Future treatment research should focus on the microbiome’s role, the use of organoids for personalised treatment, and novel therapies such as targeted therapies and immunotherapies.

## 4. Discussion

In the management of tongue cancer, the adoption of a multidisciplinary approach signifies a pivotal shift towards enhancing patient care and outcomes. This collaborative strategy underscores the integration of diverse medical specialties and allied health professions, each imparting a unique spectrum of expertise and perspectives critical to the comprehensive care of patients. The inclusion of epidemiology and aetiology specialists enriches the team’s capacity to delineate risk factors and identify populations at heightened risk, thereby informing targeted screening and referral processes that are crucial for the early detection and prompt intervention for tongue cancer.

The critical roles of nursing and allied health professionals in patient and family education on the importance of early detection highlight the multidimensional aspects of this approach. Furthermore, the concerted efforts in diagnosis, staging, and treatment planning underscore the indispensability of a collective approach, incorporating the skills of radiologists, pathologists, and surgical specialists. This collaborative ethos extends to the support provided by nursing and allied health professionals throughout diagnostic procedures and the meticulous coordination orchestrated by the surgical head and neck nurse coordinator to ensure seamless integration across specialties.

The dynamic landscape of surgical innovation in tongue cancer treatment exemplifies the critical contribution of surgical teams, augmented by the pivotal roles of nursing and allied health professionals in preparing patients for surgery and facilitating post-operative recovery. The emphasis on reconstruction and functional outcomes further illuminates the integral role of rehabilitative therapies in restoring patient function and quality of life, supported by a dedicated team of professionals focused on speech, swallowing, and physical rehabilitation.

The discourse extends to the significance of systemic therapies, including chemotherapy and immunotherapy, where the nursing and allied health cadre are instrumental in managing treatment side effects. This comprehensive care continuum necessitates a sustained focus on post-treatment recovery and long-term care, ensuring that patients receive the requisite support for managing side effects and adapting to life post-treatment.

Furthermore, the dialogue on survivorship and surveillance highlights the ongoing commitment to monitoring and supporting patients beyond the immediate treatment phase, underscoring the role of regular follow-ups in managing long-term effects and ensuring continuous patient support. The discussion also addresses the management of recurrent, residual, or metastatic disease, emphasising the essential role of palliative and supportive care in enhancing the quality of life for those facing advanced disease stages.

This study reflects the essence of a multidisciplinary approach in fostering a collaborative, learning-driven environment among healthcare professionals. It underscores the necessity for continuous engagement with the latest advances in the field, ensuring that all team members are equipped to deliver optimal care. The collective endeavour of the multidisciplinary team in navigating the complexities of tongue cancer management exemplifies a patient-centred approach that is paramount to achieving superior outcomes.

## 5. Conclusions

The effectiveness of managing tongue cancer lies in a multidisciplinary approach, integrating the expertise of medical specialties and allied health professionals. This collaboration spans from early detection and diagnosis through to post-treatment recovery and survivorship, ensuring comprehensive patient care. The key roles include epidemiologists for risk assessment, surgeons and radiologists for treatment planning, and nursing and allied health professionals for rehabilitation and supportive care. This approach is crucial in addressing not only the clinical aspects of the disease but also the patient’s QoL, especially in managing long-term effects and potential recurrences. Ultimately, this collaborative strategy aims to provide the most effective patient-centred care for those affected by tongue cancer.

## Figures and Tables

**Figure 1 cancers-16-01277-f001:**
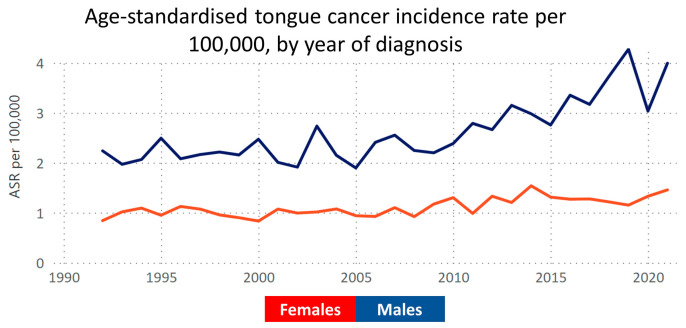
Illustration of the age-standardised incidence rates of tongue cancer over the past 30 years, utilising data from the Victorian Cancer Registry. The graph depicts the age-standardised rate (ASR) of tongue cancer incidence, demonstrating an increasing incidence indicating newly identified cases. Data accessible at https://www.cancervic.org.au/research/vcr (accessed on 2 April 2023).

**Figure 2 cancers-16-01277-f002:**
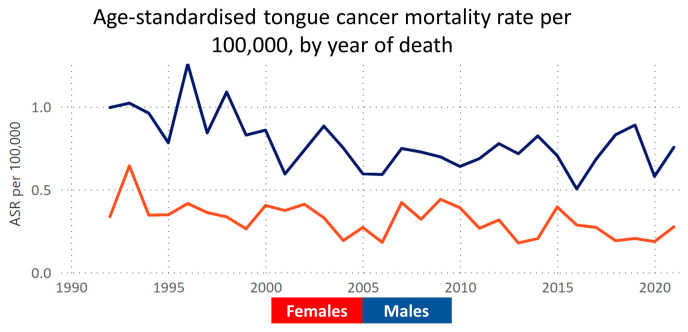
Illustration of the age-standardised mortality rates of tongue cancer over the past 30 years, utilising data from the Victorian Cancer Registry. The graph depicts the age-standardised rate (ASR) of tongue cancer mortality, showing a decrease in the rate of deaths attributed to this disease. Data accessible at https://www.cancervic.org.au/research/vcr (accessed on 2 April 2023).

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
