# Peer review of "Optimising Patient Outcomes in Tongue Cancer: A Multidisciplinary Approach"

_cancers, 2024, doi:10.3390/cancers16071277_

Round 1
Reviewer 1 Report
Comments and Suggestions for Authors
A very helpful contribution and information for those who care for head and neck cancer patients, and specifically tongue cancer. Under abstract you mentioned the impact of the symposia in different groups, how was that determined? Otherwise, you may have to clarify that the outcome of this intervention has yet to be determined.
Author Response
Thank you for your constructive feedback regarding the presentation of the impact of our symposia within the abstract. We appreciate your insight and acknowledge the necessity for clarity on how the outcomes were determined. To address your concerns, we have revised the abstract to more accurately reflect the prospective nature of the symposium's impact. It now reads: 'This initiative should be relevant to healthcare professionals, researchers, and policymakers striving to enhance patient outcomes in tongue cancer care through innovative collaboration.' We believe this amendment aligns with your guidance and better communicates our intentions. Thank you again for your valuable input.
Reviewer 2 Report
Comments and Suggestions for Authors
It is a good summary of the multidisciplinary treatment of tongue cancer. This is a useful review for those involved in the treatment of tongue cancer.
Author Response
We sincerely appreciate your encouraging comments on our summary of the multidisciplinary treatment of tongue cancer. It is gratifying to know that our review is considered useful for professionals involved in tongue cancer treatment. Your positive feedback reinforces our commitment to contributing valuable insights to the field. Thank you for recognising the efforts put into this work
Reviewer 3 Report
Comments and Suggestions for Authors
The paper is well written and structured.
1. Main Question Addressed by the Research:
The paper seeks to shed light on the new evidence regarding the optimal management of patients with tongue cancer.
2. Originality and Relevance for the Field:
While the paper does not present novel research findings, its contribution lies in its synthesis and organization of existing literature. By consolidating recent evidence, the paper oDers a valuable resource for clinicians and researchers seeking to stay updated on advancements in tongue cancer management. This review serves as a comprehensive overview of the current state of knowledge in the field, highlighting emerging trends and areas for further investigation.
3. Contribution to the Subject Area:
Compared to other reviews, the paper specifically emphasizes the impact of multidisciplinary management of the pathology, thereby enriching the discourse within the field.
4. Methodology Improvements:
From a methodological perspective, no significant improvements are necessary. However, providing more detailed information on the data extraction process from the literature would enhance clarity.
5. Consistency of Conclusions with Evidence:
The literature review is eDectively conducted and easily readable. The conclusion that a multidisciplinary approach is crucial in managing patients with tongue cancer is well-supported by the evidence presented in the paper.
6. Appropriateness of References:
The references cited in the paper are relevant and contribute to the overall coherence of the argument. By drawing on a diverse range of sources, including peer-reviewed journals and authoritative textbooks, the authors demonstrate a thorough understanding of the subject matter.
Considering that many patients needs a reconstruction with a free flap, I would add a couple of phrases regarding the functional outcomes of free flap reconstruction in oral cancer surgery. I suggest to see this: DOI: 10.1016/j.oraloncology.2023.106415
7. Additional Comments on Tables, Figures, and Data Quality:
Regarding the figures, it would be beneficial to specify what "ASR" stands for to enhance reader understanding.
Author Response
We greatly appreciate the thorough review and insightful feedback provided on our manuscript. Below, we address each of the points raised:
-
Main Question Addressed by the Research: We are pleased to clarify that our research aims to illuminate the latest evidence on the optimal management strategies for patients with tongue cancer. Thank you for recognising the importance of this question.
-
Originality and Relevance for the Field: We acknowledge your observation regarding the originality of our research. Our intention was to synthesise and organise existing literature to offer a valuable resource for those in the field, and we are glad to see this contribution recognised.
-
Contribution to the Subject Area: Thank you for appreciating our emphasis on the impact of multidisciplinary management of tongue cancer. We believe this enriches the discourse within the field and are gratified that you share this view.
-
Methodology Improvements: We appreciate your suggestion for improving the clarity of our methodology. To address this, we have added a detailed paragraph (now included in line 110 of the Materials and Methods section) that describes our literature review and data extraction process. We hope this enhancement meets your expectations.
-
Consistency of Conclusions with Evidence: We are thankful for your recognition of our literature review's effectiveness and readability. Your acknowledgment that our conclusions about the importance of a multidisciplinary approach are well-supported by evidence is highly encouraging.
-
Appropriateness of References: Following your suggestion, we have added a reference (now in line 317) that provides insights into the functional outcomes of free flap reconstruction in oral cancer surgery. This addition further enriches our discussion on the subject.
-
Additional Comments on Tables, Figures, and Data Quality: To clarify the acronym "ASR" used in our figures, we have included its definition (age-standardised rate) in the figure legends. We hope this resolves any potential confusion for readers.
We are grateful for the opportunity to enhance our manuscript based on your feedback and believe these modifications have significantly improved the clarity and quality of our work. Thank you once again for your valuable input.